# Antimicrobial Resistance of *Staphylococcus* sp. Isolated from Cheeses

**DOI:** 10.3390/ani12010036

**Published:** 2021-12-24

**Authors:** Jana Výrostková, Ivana Regecová, František Zigo, Boris Semjon, Gabriela Gregová

**Affiliations:** 1Department of Food Hygiene, Technology and Safety, The University of Veterinary Medicine and Pharmacy in Košice, Komenského 73, 041 81 Kosice, Slovakia; jana.vyrostkova@uvlf.sk (J.V.); boris.semjon@uvlf.sk (B.S.); 2Department of Animal Nutrition and Husbandry, The University of Veterinary Medicine and Pharmacy in Košice, Komenského 73, 041 81 Kosice, Slovakia; frantisek.zigo@uvlf.sk; 3Department of Public Veterinary Medicine and Animal Welfare, The University of Veterinary Medicine and Pharmacy in Košice, Komenského 73, 041 81 Kosice, Slovakia; gabriela.gregova@uvlf.sk

**Keywords:** antimicrobial resistance, MALDI-TOF MS, *mecA* gene, PCR, *Staphylococcus* sp.

## Abstract

**Simple Summary:**

As far as it is known, studies dealing with antimicrobial resistance in certain species of staphylococci, in particular, *S. chromogenes* and *S. simulans*, isolated from products made from unpasteurized milk are limited. In addition to that, little attention was paid to the resistance of staphylococcal isolates from regional sheep and goat cheeses. At this level, works are published that focus on the evaluation of resistance from only one sheep product, Bryndza. Other studies are only focused on the evaluation of resistance from raw sheep’s or goat’s milk. The study contributes to the knowledge of the possible spread of antimicrobial resistance from the farm to the final consumer in this area.

**Abstract:**

*S. aureus* and some species of coagulase-negative staphylococci, including *S. chromogenes and S. simulans*, commonly cause intramammary infections. However, little attention was paid to the antimicrobial resistance of these species with respect to their occurrence in dairy products, for example, popular sheep and goat cheeses made from unpasteurized milk. The aim of this study was to investigate such sheep and goat cheeses for the occurrence and antimicrobial resistance of the relevant staphylococci species. The staphylococcal isolates were identified by polymerase chain reaction (130 isolates) and matrix assisted laser desorption/ionization time-of-flight mass spectrometry. The most common species of *S. aureus* (56 isolates) were identified, as well as *S. chromogenes* (16 isolates) and *S. simulans* (10 isolates). Antimicrobial resistance to penicillin, oxacilin, ceftaroline, teicoplanin, gentamicin, erythromycin, tetracycline and ofloxacin was subsequently determined in these species using the agar dilution method. The highest resistance was confirmed in all species, especially to penicillin (91%) and erythromycin (67%). The highest sensitivity was confirmed to ofloxacin (83%). Due to the high incidence of penicillin and oxacilin-resistant staphylococci, the *mecA* gene was detected by polymerase chain reaction, which was confirmed only in *S. aureus* isolates (19%). Our study shows that the tested strains (77%) were resistant to more than one antibiotic at a time.

## 1. Introduction

One of the oldest fermented foods is cheese [1,2]. Cheese has been part of the human diet for thousands of years, although there have been changes in dietary patterns associated with technical, social, and economic progress in individual geographical regions. The cheese fermentation process and use depend considerably on culture and tradition [3]. There are numerous variations in the characteristics in cheese, including texture, aroma, visual presentation, and flavour, that depend on the activity of microorganisms and the cheesemaking process [4].

In addition to the beneficial microflora, undesirable microorganisms are also present in the cheeses. Among them, we also include bacteria of the genus *Staphylococcus* sp., which are the main causative agent of mastitis, with a higher prevalence in cases of clinical and subclinical manifestations. Recently, in addition to *S. aureus*, *S. chromogenes* and *S. simulans* species have also often been detected in mastitis [5,6,7,8].

The presence of antimicrobial resistance has recently been confirmed in the above-mentioned staphylococcal species, which is an increasing problem, so the collection of information on pathogen resistance is very important from an epidemiological point of view. Especially, a serious risk is associated with multiresistant strains and their resistance to more than one antibiotic [9]. Recent studies also suggest the presence of methicillin-resistant staphylococci (MRS) that have been identified in unpasteurized milk and dairy products, including cheeses. The opportunistic ability of MRS strains to cause mastitis and serve as a source of zoonotic infections is a considerable threat to public health. Moreover, they present a reservoir of antimicrobial resistance genes on dairy farms. The most widely reported MRS species is *Staphylococcus aureus* (MRSA). Coagulase-negative staphylococci (CNS) were also identified as MRS isolates [10].

At the same time, the degree of resistance to different antibiotics varies from species of staphylococci; therefore, their identification of them is important. Traditional methods used to identify and classify bacteria are based on the analysis of morphological, physiological and biochemical traits or genetic approaches (DNA-DNA or RNA-DNA hybridization, determination of G+C content in DNA). Those methods are currently supplemented by PCR methods to analyze the sequence of small rRNA subunits [11].

Matrix-assisted laser desorption ionization time-of-flight mass spectrometry (MALDI-TOF MS) currently serves as a method suitable for this purpose. Recently, considerable improvement has been made in MALDI-TOF MS tools intended for microbiological identification of potential pathogens and foodborne pathogens. These commercial, MALDI-TOF MS devices are convenient and utilize their own algorithms and databases Many authors described them as fast, cost-effective systems with an accurate and reliable performance [12,13].

Based on the abovementioned, the aim of our study is to detect the presence of *Staphylococcus* sp. by PCR and MALDI-TOF mass spectroscopy of sheep and goat cheese produced on farms in Slovakia and, subsequently, determine their antimicrobial resistance. The cheeses tested by us belong to the group of so-called “ready to-it” made from raw milk with no special starter culture [14].

## 2. Materials and Methods

Staphylococci were isolated from cheese samples (10 sheep cheese samples and 10 goat cheese samples) taken between May and August 2021. Cheeses were obtained from a farm located in the border area of Slovakia and Hungary in the region Slanské vrchy. This farm is producing milk and milk products in accordance with legal requirements in the EU and is an officially approved establishment. The farm raises Valachian sheep and white shorthaired goats, which were grazed in a wild pasture. The cheeses were made from non-pasteurized sheep and goat milk without adding a cheese starter culture and were ripened for 30 days.

### 2.1. General Microbiological Analysis

Preparation of test samples of cheeses and decimal dilutions were prepared according to STN EN ISO 6887-5 (2011) [15] and, subsequently, for the tested samples, the total viable count (TVC) according to STN EN ISO 4833-1 (2013) using Plate Count Agar. After incubating the inoculated plates for 72 h at 30 °C, the inoculated plates where more than 10 and less than 300 colonies grew were selected for the TVC calculation.

### 2.2. Isolation of Strains

Subsequently, basic suspension and decimal dilutions were prepared from all tested samples according to ISO 6887-5 [15].

Isolates of staphylococci from the examined samples were obtained according to ISO 6888-1 [16] using Baird-Parker selective arbitration medium. The inoculated plates were incubated at 37 for 24 h. Subsequently, plates with more than 10 and less than 150 atypical and typical colonies were considered for staphylococcal counts. Based on their characteristic appearance, 2 typical colonies (1.0–1.5 mm colony, black or grey colonies with halo) and 2 atypical colonies (black or gray colonies without halo), each plate was inoculated with a sterile bacterial loop on the surface of Columbia blood agar (Oxoid Ltd., Hampshire, UK) and incubated at 37 °C for 24 to 48 h. After incubation, individual strains were used for identification by PCR and MALDI-TOF MS.

### 2.3. Identification of Staphylococcal Isolates

Total genomic DNA was isolated from staphylococcal strains as described by Hein et al. [17].

The obtained supernatant was used as a DNA source in PCR reactions. The 16S ribosomal RNA gene specific of *Staphylococcus* sp. was amplified in a thermal cycler (Techne Touchgene FTGPO2TD, Techne, Cambridge, UK). Primers used to amplify a given 16S rRNA, 16S1 (CAGCTCGTGTCGTGAGATGT) and 16S2 (AATCATTTGTCCCACCTTCG), were synthesized (Amplia s.r.o., Bratislava, Slovakia) and used according to Strommenger et al. [18]. The reaction mixture in a volume of 20 µL contained 1 µL genomic DNA, 10 pmol.L^−1^ of each primer and HotFirepol^®^ Master Mix (Amplia s.r.o., Bratislava, Slovakia). The amplification was terminated by cooling to 6 °C. The PCR protocol was as follows: initial denaturation at 95 °C for 12 min, 25 cycles consisting of denaturation at 95 °C for 20 s, annealing at 55 °C for 1 min and extension at 72 °C for 2 min. The final extension at 72 °C for 10 min followed the last cycle. PCR products (420 bp) were visualized using MiniBIS Pro^®^, (DNR Bio-Imaging System, Ltd., Jerusalem, Israel).

The species identification of bacteria was subsequently provided with the help of MALDI-TOF MS according to the standard sample preparation protocol of manual Bruker Daltonics [19]. The analysis of the results was performed in a Ultraflex III device (Bruker, Billerica, Massachusetts, USA) using Flex Analysis software, version 3.0, and evaluated with by BioTyper software, version 1.1 (Bruker, Billerica, MA, USA).

Retrospective PCR products were sequenced in species-identified *S. aureus, S. chromogenes* and *S. simulans* strains by MALDI-TOF MS. The sequencing of the PCR products was performed according to the Sanger method (GATC Biotech AG, Konstanz, Germany). The obtained strains were sent to the GenBank—EMBL database for comparison with the sequences available in the nucleotide database of the National Center for Biotechnology Information (NCBI) available at http://www.ncbi.nlm.nih.gov/BLAST (accessed on 20 November 2021).

### 2.4. Detection of Antimicrobial Resistance

The susceptibility of isolated bacterial strains to selected antibiotics was determined by the agar dilution method (ADM) according to the procedure described in the CLSI document [20]. ADM is performed on Petri dishes with Müeller–Hinton agar (Hi-Media, Mumbai, India) in duplicate. Test plates containing different concentrations of antibiotics were used to determine minimum inhibitory concentrations (MICs). Turbidity of 24-hour bacterial suspensions was adjusted to the 0.5 McFarland turbidity standard. Drops of such adjusted suspensions were applied in parallel onto the surface of the test plates which were then incubated at 37 °C for 24 h. After incubation, we visually determined the minimum concentration of respective antibiotics that inhibited the growth of the investigated staphylococcal strains. The results were evaluated according to document CLSI [20].

In determining the minimal inhibitory concentrations (MIC) used in assay plates with final concentration of antibiotics on *Staphylococcus* spp.: Penicillin (PEN) 0,.12;0.25; 0.5 mg·L^−1^; Oxacillin (OX) 0.12; 0.25; 0.5; 1.0 mg·L^−1^; Ceftaroline (KF) 0.5;1.0; 2.0; 4.0; 8.0 mg·L^−1^; Teicoplanin (TEC) 4.0; 8.0; 16.0; 32.0; 64.0 mg·L^−1^; Gentamicin (GN) 2.0; 4.0; 8.0; 16.0; 32.0; 64.0 mg·L^−1^; Erythromycin (E) 0.25; 0.5; 1.0; 2.0; 4.0; 8.0; 16.0 mg·L^−1^;Tetracycline (TE) 2.0; 4.0; 8.0; 16.0; 32.0 mg·L^−1^; Ofloxacin (OFX) 0.5; 1.0; 2.0; 4.0; 8.0 mg·L^−1^.

To confirm the MRS strains, the presence of the *mecA* gene was detected by PCR according to Poulsen et al. [21]. The primers used to confirm the presence of the *mecA* gene were MecA-1 (5-GGGATCATAGCGTCATTATTC) and MecA-2 (5-AACGATTGTGACACGATAGCC) (Amplia s.r.o., Bratislava, Slovakia). The size of the amplification product using these two primers was 527 bp.

Reference strain of *S. aureus* CCM 4750 (Czech Collection of Microorganisms, Brno, Czech Republic) were used in this study as a positive control for the agar dilution method and S. aureus CCM 4223 for the polymerase chain reaction.

### 2.5. Statistical Analysis

Results obtained in this study were analyzed statistically using GraphPad Prism 5.0 software (2007). Comparison of individual proportions was made using the Chi-square test (χ^2^ test). The dependence of the individual signs was considered significant at the level of α = 0.05, with critical values = 4.776 for *Staphylococcus* sp.

## 3. Results

The total viable count and staphylococcal counts according to the relevant standards were determined by microbiological culture examination of individual cheese samples (Table 1). Subsequently, identification isolates by PCR method detected 130 isolates of *Staphylococcus* spp.

In the study, subsequently, we carried out identification of the species *S. aureus* (56 isolates), *S. chromogenes* (16 isolates) and *S. simulans* (10 isolates) by the MALDI –TOF MS method. The score value of the identified strains ranged from 2.096 to 2.268 in *S. aureus*, between 2.076 and 2.105 in *S. chromogenes* and from 2.002 to 2.224 in *S. simulans*.

A total of 82 *Staphylococcus* isolates, identified by MALDI-TOF MS, were subjected to partial 16S rRNA gene sequencing. All PCR products showed the expected size (420 bp). The 16S rRNA sequences were compared with the sequences deposited in GenBank. Sequence similarity values greater than 99% (≥99%) were considered reliable according to the CLSI (2018). In all cases, the sequence similarities with GenBank sequences ranged from 99 to 100%. The specific results of the partial sequencing of 16S rRNA and MALDI-TOF MS identification are shown in Table 2.

Because goat and sheep milk products are a good substrate for the growth of resistant staphylococci, we also examined the identified isolates for their resistance to selected antibiotics. Bacteria of *Staphylococcus* spp. (82 isolates) showed the highest resistance to penicillin (98%; 80 isolates) (Table 3).

In particular, resistance to penicillin (100%; 16 isolates), oxacilin (94%; 15 isolates), erythromycin (63%; 10 isolates) and tetracycline (75%; 12 isolates) was detected in *S. chromogenes*. A 100% sensitivity has only been reported for teicoplanin in these species. *S. simulans* strains, similar to *S. chromogenes*, showed 100% resistance to erythromycin, but no strain was resistant to tetracycline, teicoplanin and ofloxacin. *S. aureus* was also confirmed to have the highest resistance to penicillin (96%; 54 isolates) and oxacilin (98%; 53 isolates) than *S. simulans* and *S. chromogenes*. In this species, each strain was detected to be resistant to at least one antibiotic. Intermediate susceptibility in *S. aureus* was confirmed in 19% of isolates against ceftaroline, teicoplanin, erythromycin and 43% of isolates against gentamicin. The occurrence of resistant strains for each type of cheese is listed in Table 4.

Based on the phenotypic manifestation of antimicrobial resistance, the presence of MRS strains was assumed, which we confirmed by PCR detection of the *mecA* gene. The presence of strains with the *mecA* gene was detected. All positive strains belonged to *S. aureus* (19%; 10 isolates). In *S. chromogenes* and *S. simulans*, *mecA* gene occurrence was not confirmed by PCR.

Multiresistance was also confirmed in the examined strains (Figure 1) in *S. aureus* in 73%, *S. simulans* (80%) and *S. chromogenes* (87%). Most often, resistance to two antibiotics simultaneously (PEN-E) was detected. Multidrug resistance to six antibiotics (PEN-KF-E-GN-TE-OFX) was also confirmed in one strain of *S. aureus*.

## 4. Discussion

The basic requirement of the entire food chain is food safety. Each part of the chain is responsible for safe food. In the Dolezalova et al. [22] study, they describe the total number of microorganisms as a general indicator and the examined cheese samples ranged up to 10^8^ CFU/g. Regulation (EC) No. 2073/2005 [23] and Government Regulation (EC) No. 312/2003 [24] lays down production limits for raw sheep’s and goat’s milk but not specifically for dairy products produced from raw sheep’s milk. However, the TVC values determined in the cheeses indicate a possible contamination during the production of such cheeses, which must be eliminated in the production process by the correct setting of the HACCP system in the operation of Carrascosa et al. [25]. Higher TVC in cheese samples was also observed in our study. This may be due to the ripening temperature of the cheeses at 30 °C, which is favorable for the growth of beneficial microflora but also for contaminating microflora [26]. Similarly, higher TVC values in cheese samples were recorded by Carrascosa et al. [25].

Microbiological examination of cheese samples and subsequent identification of bacteria by PCR confirmed the presence of bacteria of the genus *Staphylococcus* spp. The 16S rRNA gene serves as an excellent target for most staphylococci. For some coagulase-negative staphylococci, the separation between species can be difficult due to the lack of sufficient heterogeneity within the 16S rRNA gene [27]. The obtained bacteria were subjected to species identification by MALDI-TOF MS, which confirmed the presence of *S. aureus*, *S. simulans*, *S. chromogenes*. Chen et al., Deng et al. and Cheng et al. [28,29,30] claimed that a score value of 2.000–2.300 in the MALDI-TOF MS indicated a highly probable identification of the species, a score value in the range of 1.7–1.999 was considered an identification of the genus and probable identification of the species, while a score value of 1.699–0.000 was considered a reliable identification. Based on these facts, we can confirm the exact species identification of isolates tested by us.

Similar confirmation was reported by Prod’hom et al. [31]. These authors used blood culture to propagate staphylococci that were then identified by MALDI-TOF MS. In this way, they were able to identify 25 isolates of *S. aureus*. Other groups have also confirmed the high efficiency of staphylococci identification at the species level only using only MALDI-TOF MS [32,33,34]. Clerc et al. [35] also identified coagulase-positive staphylococci by MALDI-TOF MS with a high score over 2.000.

Since the bacterium of the genus *Staphylococcus* has recently shown antimicrobial resistance, in our studies, to proceed to the detection of strains identified by us, it was detected in up to 64% of isolates. Similarly, Jamali et al. [36] reported that staphylococci isolated from raw milk and milk products was highly resistant to tetracyclin and penicillin but susceptible to oxacillin, lincomycin, klindamycin, erythromycin, streptomycin, cefoxitin, kanamycin, gentamycin and chloramphenicol. Kraemer et al. [37] confirmed a more frequent resistance to erythromycin compared to other antimicrobial agents. The agar dilution method showed that isolates of the *Staphylococcus* spp. resistance to macrolides (erythromycin) was higher (75%; 60 isolates).

Resistance to β-lactam antibiotics in staphylococcal isolates was also confirmed by Sampimon [38] who found that resistance to penicillin was 18% in *S. chromogenes*. Similar proportions of *β*-lactamase production for *S. chromogenes* (18%) were also reported in a US study [39].

Persson-Waller et al. [40] also confirmed resistance to β-lactam antibiotics in *S. chromogenes* (33%) in their study.

Erythromycin resistance was also detected in *S. chromogenes* and *S. simulans* by Lüthje and Schwarz [41]. Erythromycin resistance in food isolates of *S. simulans* was also confirmed by Chajecka-Wierzchowska et al. [42]. At the same time, they confirmed tetracycline resistance in these isolates.

The lowest resistance to teicoplanin was confirmed. Vasiľ et al. [43] confirmed a higher resistance to erythromycin (12%) in *S. aureus*. Intermediate susceptibility of erythromycin was confirmed in 14 isolates. For other antibiotics, the incidence of resistant strains of *S. aureus* was relatively low.

Overall, common resistance to β-lactam antibiotics and the presence of the *mecA* gene were found in our study, similar to the study by Rajala-Schultz et al. [44] and Persson-Waller et al. [40] who tested the staphylococcal strains in raw milk and mastitis.

The prevalence of the presence of the *mecA* gene varied markedly between CNS species and was significantly higher in *S. epidermidis* and *S. haemolyticus* (∼40%) than in *S. simulans* and *S. chromogenes*. Originally, methicillin-resistant *S. aureus* (MRSA) were detected primarily in humans, only later they were found also in animals [45]. Recently, the increase in resistance to methicillin, penicillin and oxacillin of staphylococci strains constitutes a serious clinical and epidemiological issue. The presence of methicillin-resistant *S. aureus* in milk is not considered a serious food safety issue as milk is commonly heat treated before consumption. However, there are also exceptions involving the consumption of raw milk by farmers and making milk products from unpasteurized milk [46]. Such practice may expose people to MRSA. Recent reports revealed that MRSA was also associated with cases of bovine and caprine mastitis [47,48]. Bogdanovičová et al. [49] also confirmed the presence of the *mecA* gene in 9.7% of the examined *S. aureus* strains examined.

In our study, multidrug resistance was also detected in the isolates. Resistance to more than one antibiotic was also confirmed by Persson-Waller et al. [40], where multidrug resistance occurred in 9% of a total of 56 staphylococcal isolates. Nunes et al. [50] reported that out of 19 CNS strains, 14 showed multiresistance to antimicrobial agents. The authors detected resistance to β-lactams (oxacillin, penicillin and/or cefoxitin) and to vancomycin in 73% of the total CNS strains identified. Resistance to tetracycline and gentamicin was confirmed in nine strains (64%), neomycin, erythromycin and chloramphenicol in eight strains (57%), sulfamethoprim in seven strains (50%), linezolid in five strains (36%), rifampicin in three strains (21%), ciprofloxacin and cefepime in two strains (14%) and, finally, only one strain (7%) was resistant to clindamycin.

Recently, attention has been focused mainly on resistance to penicillin-stable penicillins, which is referred to as “methicillin resistance” or “oxacillin resistance”. Most resistance to methicillin (oxacillin) is mediated by *mecA* encoding PBP2a. Isolates, in which the presence of the *mecA* gene is confirmed, should be classified as resistant to methicillin (oxacillin) according to CLSI [23].

The multiresistance of CNS strains reported by the above authors is consistent with the observations obtained in the previous studies on coagulase-negative and coagulase-positive staphylococci that revealed the presence of several resistant and multiresistant staphylococcal strains in raw milk and products from unpasteurized milk [51].

Our results are supported by the study by Seng et al. [52] who reported as high as 80% resistance to several types of antibiotics in staphylococci isolates. The study by Hleba et al. [53] also reported the presence of multiresistant bacteria in milk and milk products.

It is common belief that the increased resistance to antimicrobials is associated with the extensive and sometimes even unreasonable use of these drugs in human and veterinary medicine [54].

## 5. Conclusions

Investigation of selected milk products brought new information on the prevalence of resistant, multi-resistant staphylococci and MRSA strains in sheep and goat cheeses produced from unpasteurized milk in Slovakia. Our study also indicated the need for rapid and effective identification of individual staphylococcal species because of differences in antimicrobial profiles of individual species and their relevance from the point of view of protecting the health of humans and animals.

## Figures and Tables

**Figure 1 animals-12-00036-f001:**
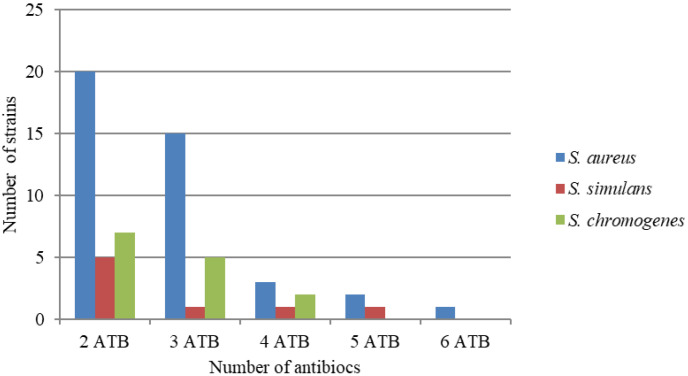
Number of multiresistance staphylococcal strains. ATB—antibiotics.

**Table 1 animals-12-00036-t001:** Total viable count and count of staphylococci from sheep and goat cheese (log CFU/g).

Type of Samples	Statistical Value	Total Viable Count	*Staphylococcus* sp.
Goat’s cheese (*n* = 10)	Minimum	6.8	3.2
	Maximum	8.3	4.2
	Mean ± SD	7.9 ± 0.4	3.8 ± 0.3
Sheep’s cheese (*n* = 10)	Minimum	6.9	3.3
	Maximum	8.1	3.9
	Mean ± SD	7.7 ± 0.4	3.6 ± 0.2

*n*—number of sample.

**Table 2 animals-12-00036-t002:** Identification of 82 *Staphylococcus* isolates from cheese through 16S rRNA sequencing and MALDI-TOF MS.

Number of Isolates	16S rRNA Sequencing Result	Accession Numbers in GenBank	Sequence Similarity %	MALDI-TOF MS	Score Value
14	*S. aureus*	OK285211.1	100	*S. aureus*	2.097–2.268
10	*S. aureus*	CP084892.1	100	*S. aureus*	2.215–2.268
9	*S. aureus*	OL344097.1	100	*S. aureus*	2.096–2.198
5	*S. aureus*	AP025177.1	99	*S. aureus*	2.098–2.215
4	*S. aureus*	CP021178.1	100	*S. aureus*	2.126–2.214
4	*S. aureus*	CP084107.1	100	*S. aureus*	2.115–2.229
3	*S. aureus*	OL336429.1	100	*S. aureus*	2.098–2.183
2	*S. aureus*	CP084878.1	100	*S. aureus*	2.176–2.204
2	*S. aureus*	AP025176.1	99	*S. aureus*	2.096–2.145
2	*S. aureus*	OK576712.1	99	*S. aureus*	2.115–2.178
1	*S. aureus*	OL345568.1	100	*S. aureus*	2.099
3	*S. simulans*	MK015778.1	100	*S. simulans*	2.100–2.224
3	*S. simulans*	NR_036906.1	100	*S. simulans*	2.891–2.189
1	*S. simulans*	MF678910.1	99	*S. simulans*	2.145
1	*S. simulans*	FN646077.1	99	*S. simulans*	2.058
1	*S. simulans*	LC437030.1	99	*S. simulans*	2.002
1	*S. simulans*	KC849411.1	100	*S. simulans*	2.220
5	*S. chromogenes*	MT913000.1	99	*S. chromogenes*	2.076–2.098
4	*S. chromogenes*	CP031471.1	100	*S. chromogenes*	2.097–2.105
2	*S. chromogenes*	CP031274.1	100	*S. chromogenes*	2.102
2	*S. chromogenes*	CP046028.1	100	*S. chromogenes*	2.088–2.100
2	*S. chromogenes*	CP031470.1	100	*S. chromogenes*	2.085–2.101
1	*S. chromogenes*	JN426805.1	100	*S. chromogenes*	2.099

**Table 3 animals-12-00036-t003:** Number of susceptible (S), intermediately susceptible (IS) and resistant (R) species of *Staphylococcus* spp.

Strains		Antibiotics								
			PEN	OX	KF	TEC	GN	E	TE	OFX
*S. aureus*	(*n* = 56)	S	2	3	40	42	22	6	44	48
		IS	0	0	10	10	24	10	2	2
		R	54	53	6	4	10	40	10	6
*S. simulans*	(*n* = 10)	S	0	1	4	10	4	0	10	10
		IS	0	0	2	0	4	0	0	0
		R	10	10	4	0	2	10	0	0
*S. chromogenes*	(*n* = 16)	S	0	1	8	6	10	2	4	10
		IS	0	0	4	10	4	4	0	2
		R	16	15	4	0	2	10	12	4
Chi-quadrate test		G	25.42 ^1^	22.32 ^1^	4.708	0.964	2.862	19.22 ^1^	6.42 ^1^	4.632

Chi-quadrate test (significance level α = 0.05; critical value χ^2^ = 4.776; G—testing value); the independence of the individual characters at the significance level α =0.05 has not been rejected, i.e., statistical independence was confirmed when G < χ^2^; ^1^—the independence of individual characters (S; IS, R) at the significance level α = 0.05 was rejected when G ˃ χ^2^, i.e., the statistically confirmed dependence of the observed traits in the test group of S. aureus (*n* = 56); *S. simulans* (*n* = 10) and S. chromogenes (*n* = 16) against antibiotic substances with PEN, E and TE.

**Table 4 animals-12-00036-t004:** Number of resistant isolates isolated from goat and sheep cheese.

		*n*	PEN	OX	KF	TEC	GN	E	TE	OFX
Goat cheese	*S. aureus*	29	28	27	2	2	3	19	5	4
	*S. simulans*	6	6	6	1	0	1	6	0	0
	*S. chromogenes*	7	7	6	2	0	2	5	5	1
Sheep cheese	*S. aureus*	27	26	26	4	2	7	21	5	2
	*S. simulans*	4	4	4	3	0	1	4	0	0
	*S. chromogenes*	9	9	9	2	0	0	5	7	3

*n*—total number of strains.

## Data Availability

The study did not report any data.

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
