# Peer review of "Antimicrobial Resistance of Staphylococcus sp. Isolated from Cheeses"

_animals, 2021, doi:10.3390/ani12010036_

Round 1

Reviewer 1 Report

This manuscript presents data regarding the antibiotic resistance of Staphylococcus spp isolated from sheep and goat cheese. Authors should significantly improve the manuscript before it could be reconsidered for publication according to the following:

  1. Editing of English language and correction of typing errors  is essential.
  2. Methods are not adequately described. Lines 108-111: According to authors bacterial isolates where identified as Staphylococcus spp by PCR. All 130 isolates were tested by PCR? All amplicons sequenced? Obtained sequences were analyzed by BLAST at NCBI site. Authors should cite BLAST publication  and they should describe in detail the  selected BLAST parameters   in  their analysis,
  3.  Result presentation should be improved. For instance Fig.1 and Fig 2 should be omitted. This kind of agarose gel pictures are not anymore presented in scientific literature. Instead, it could be very useful a table presenting the data of BLAST analysis, thus confirming the identification of bacterial isolates.
  4. I have some questions and suggestions regarding the design of this study. Why authors did not check methicilin resistance of S. aureus isolates phenotypically e.g. by ADM? The presence of mecA  is not an evidence of methicilin resistance in all cases.  Actually I would find it  more interesting if authors could provide data regarding the enterotoxicity of identified S. aureus isolates (f.i. detection of diverse genes encoding enterotoxins).

Author Response

Ivana Regecová, DVM, PhD.

Department of Food Hygiene, Technology and Safety

University of Veterinary Medicine and Pharmacy

Komenského 73

041 81 Košice

Slovak Republic

E-mail: ivana.regecova@uvlf.sk

December 08. 2021

Editorial Board

Animals

MDPI

St. Alban-Anlage 66, 4052 Basel,

Switzerland

Dear Reviewer

Please find attached our revised research article „ Antimicrobial Resistance of Staphylococcus sp. Isolated from cheeses “written by Jana Výrostková, Ivana Regecová*, František Zigo, Boris Semjon and Gabriela Gregová which we would like to submit for consideration to the Animals in special issue " Nutritional Quality Assessment in Milk and Dairy Products".

We would like to thank the reviewer for comments making this manuscript clearer and more reliable. All recommendations have been accepted by the authors. English language has been revised and errors have been corrected.  All edits are marked in the document via the „Track Changes“ function. Thank you for considering this manuscript.

Yours Sincerely,

Ivana Regecová, DVM, PhD.

Reviewer 2 Report

The paper is about the isolation and characterization of Staphylococcus from raw milk cheeses. Antimicrobial resistance is a serious concern in the EU; thus any research on this field is acknowledge. This said, the manuscript offers very preliminary results and more work is needed to merit publication.

Typing of MRSA by MLST and SCCmec is simple and informative and allows comparison of the results with those obtained in other locations.

Other comments are provided below:

  • Please, include the number of isolates in the abstract
  • L81-84. Include information about cheese samples. Where they purchased in markets? Where they in accordance with legal requirements in the EU?. Taking into account the short ripening timeDid the authors carry out general microbiological analysis (total bacterial counts)?
  • L85-92. I understand that the medium was BP with egg yolk tellurite; if so, what was the  criterium to select the colonies?. Colonies with clear halo?. Did the authors carry out staphylococcus count? . Please explain.
  • L128-133. The selection of antibiotics seems to follow no rationale, surely not the CLSI recommendation for phenotypic detection of MRS.
  • Presentation of results must be improved. Please include microbiological results (Staph counts, number of isolates per cheese type...)
  • L148-150. The authors do not mention sequencing results. Where the results in accordance with MALDI-TOF identification.
  • Figure 2. Are the authors sure that reference strain is OK?. According to CCM catalogue, strain CCM 4223 is methicillin susceptible.
  • The discussion section must be rewritten taking into consideration the previous comments.

Author Response

(The authors gave the same response as above.)

Round 2

Reviewer 1 Report

This   manuscript is significantly improved. Therefore, it can be accepted in  present form.

Author Response

Ivana Regecová, DVM, PhD.

Department of Food Hygiene, Technology and Safety

University of Veterinary Medicine and Pharmacy

Komenského 73

041 81 Košice

Slovak Republic

E-mail: ivana.regecova@uvlf.sk

December 13. 2021

Editorial Board

Animals

MDPI

St. Alban-Anlage 66, 4052 Basel,

Switzerland

Dear Reviewer

Please find attached our revised research article „Antimicrobial Resistance of Staphylococcus sp. Isolated from cheeses “written by Jana Výrostková, Ivana Regecová*, František Zigo, Boris Semjon and Gabriela Gregová which we would like to submit for consideration to the Animals in special issue " Nutritional Quality Assessment in Milk and Dairy Products".

We would like to thank the reviewer for the comments that make this manuscript clearer and more reliable. All recommendations were accepted by the authors. All edits are marked in the document using the "Track Changes" feature.Thank you for considering this manuscript.

Yours Sincerely,

Ivana Regecová, DVM, PhD.

Reviewer 2 Report

The authors have improved the manuscript and now it's acceptable, though some minor changes need to be made.

  • L172. Table 1. Please indicate units (log CFU/ml??)
  • L235. KTJ/ml should be CFU/ml
  • The results of TVC are higher than the maximum level admitted in the EU for milk of species other than cow destined to production of milk products without thermal treatment. The authors should discuss this point.

Author Response

(The authors gave the same response as above.)
